

# A global inventory of historical documentary evidence related to climate since the 15th century

Angela-Maria Burgdorf [1,2]

[1]Oeschger Centre for Climate Change Research, University of Bern, Bern, 3012, Switzerland
[2]Institute of Geography, University of Bern, Bern, 3012, Switzerland

*Correspondence to*: Angela-Maria Burgdorf (angela-maria.burgdorf@giub.unibe.ch)

**Abstract.** Climatic variations have impacted societies since the very beginning of human history. In order to keep track of climatic changes over time, humans have thus often closely monitored the weather as well as natural phenomena influencing everyday life. Resulting documentary evidence from archives of societies enables invaluable insights into the past climate beyond the timescale of instrumental and early instrumental measurements. This information complements other proxies from archives of nature such as tree rings in climate reconstructions, as documentary evidence often covers seasons (e.g., winter) and regions (e.g., Africa, Western Russia, and Siberia, China) that are not well covered with natural proxies. While a mature body of research on detecting climate signals from historical documents exists, the large majority of studies is confined to a local or regional scale and thus lacks a global perspective. Moreover, many studies from before the 1980s have not made the transition into the digital age and, hence, are essentially forgotten. Here, I attempt to compile the first-ever systematic global inventory of documentary evidence related to climate extending back to the Late Medieval Period. It combines information on past climate from all around the world, retrieved from many studies on historical documentary sources. Historical evidence range from personal diaries, chronicles, administrative/ clerical documents to ship logbooks and newspaper articles. They include records of many sorts, e.g., tithes records, rogation ceremonies, extreme events like droughts and floods, as well as weather and phenological observations. The inventory, published as an electronic supplement, comprises detailed event chronologies, time series, proxy indices, and calibrated reconstructions, with the majority of the documentary records providing indications on past temperature and precipitation anomalies. The overall focus is on document-based time series with significant potential for climate reconstruction. For each included record series, extensive meta information and directions to the data (if available) are given. To highlight the potential of documentary data for climate science three case studies are presented and evaluated with different global reanalysis products.

This comprehensive inventory promotes the first-ever global perspective on historical documentary climate records and, thus, lies the foundation for incorporating historical documentary evidence into climate reconstruction on a global scale, complementing early instrumental measurements as well as natural climate proxies.



# 1 Introduction

To reconstruct past climate and its variability we rely on information from multiple archives. While the weather, climatic variations, and extreme events have been observed and documented since the classical epoch, it was not until the nineteenth century that meteorological instrumental measurement networks were established on national levels all over the world (Brönnimann, 2015). Before that, early instrumental measurements from individuals and meteorological networks by private initiatives recorded information on weather and climate and allow insights into climate variations on a daily to decadal scale as far back as 1658 for Paris (Rousseau, 2015) and 1659 for Central England (Manley, 1974; Parker et al., 1992). See Brönnimann et al. (2019) for a comprehensive inventory of metadata on globally available early instrumental data before 1850. Going further back in time, climate information with annual or seasonal resolution can be deduced from either natural proxies such as tree rings, ice cores, corals, varve sediments or from historical documentary evidence from archives of societies. Climate reconstruction based on natural proxies is a well-established and widely accepted approach, compellingly presented in many studies (e.g., Rutherford et al., 2005; Mann et al., 2008; Phipps et al., 2013). Alternatively, climate information can be deduced from historical documentary evidence. Since the very beginning of human history climatic variations have affected societies across the globe. Humans in all different cultural settings have thus often closely monitored and documented the weather as well as natural phenomena influencing everyday life. These manmade documentary records stored in archives of societies represent records of climate variability over time and can provide important complementary information for climate reconstruction as they often cover seasons and regions that are not well represented with natural proxies. Tree-ring proxies, for example, are restricted to annual signals of temperature or precipitation during the growing season. Documentary ice freeze-up and break-up records allow for very precisely dated temperature signals for winter and spring. Nevertheless, also documentary proxies depend on interpretation and are thus to be used mindfully.

The fact that accurate seasonal, and even monthly signals can be extracted from documentary evidence from all continents make them particularly valuable when it comes to reconstructing the climate on a large scale. In their review of the climate since A.D. 1500, Bradley and Jones (1995) evaluate available paleoclimatic data for climate reconstruction and ascribe particular potential to documentary records together with tree-rings. Pfister et al. (2009) discuss the role of documentary evidence for climate reconstruction in detail, addressing its potential and caveats. The "Palgrave Handbook of Climate History" provides a comprehensive overview of the state-of-the-field on weather and climate of the past (White et al., 2018). Besides describing relevant methods, techniques, and sources, as well as discussing major findings of the field of historical climate reconstruction, a thorough overview of existing documentary sources through time and from all continents is presented.

To advance the role of historical climatology in paleoclimate reconstructions and to coordinate the global effort to establish best practices for climate reconstruction from the archives of societies, the PAGES CRIAS (Climate Reconstruction and Impacts from the Archives of Societies) working group was formed in 2018. Unlike in natural proxy communities (e.g., the tree-ring community) where global networks exist, historical documentary evidence has rarely been viewed in a large-scale context because cultural barriers have long made it difficult to facilitate a global view. To some extent, this still holds true



today, but it is part of the CRIAS working group's agenda to work towards overcoming this cultural gap. An overview of the current state-of-the-field, and the most recent accomplishments are published in the Pages Global Change Magazine (Camenisch et al., 2020).

While a mature body of research on detecting climate signals from historical documents exists, the large majority of studies and their analyses and application to climate reconstructions are confined to a local or regional scale and thus lack a global perspective. On one hand, this is due to the fact that historical documentary evidence is predominantly collected and evaluated with local or regional foci. On the other hand, global-scale paleo reconstructions mainly rely on natural proxies such as tree-rings, coral, and ice cores that are available for long timescales and many parts of the world. A prominent example is the
PAGES2k initiative. In a global effort, the PAGES2k Consortium has compiled temperature-sensitive proxy records for the Common Era into the PAGES2k temperature paleo-record database (Emile-Geay et al., 2017). Out of the 692 records included, only 14 are documentary evidence, which goes to show its secondary role in the field of climate reconstructions. A possible explanation for this might simply be the fact that even though existing, historical data is not readily available since the data itself is often not published. However, in recent years research groups from all around the globe have made a considerable
effort to publish historical documentary records related to climate. Many different databases and repositories exist online and provide access to historical documentary evidence of climatic parameters such as temperature, precipitation, droughts and floods, storms, etc. Table 1 gives an overview of the most relevant compilations, focusing on different regions and topics. Based on the available data they contain, databases and repositories can be divided into two categories: The ones that predominantly comprise raw historical documentary evidence related to climate (e.g., Tambora.org, Euro-Climhist,
REACHES, TEMPEST, CLIWOC) and the ones that provide (interpreted) documentary data time series of climatic parameters (e.g., PAGES2k, NOAA Paleo, NSIDC, JCDP, RECLIDO, Salvá Sinobas, Vareclim).

As apparent from Table 1, historical documentary data related to climate is available online but distributed on many different platforms. Furthermore, these only represent a small fraction of the existing records. The goal of this work is to compile a
comprehensive, global, multi-variable inventory, including relevant metadata, corresponding references, and directions where to find the data (if available). It shows the potential of documentary data for climate science and aims at facilitating the work of scholars and scientists studying climate-related topics.





**Table 1.** Overview of available global and national repositories and databases containing documentary evidence. $N_{all}$ depicts the total number of series available on the platform, $N_{docu}$ the number of series or databases based on historical documentary evidence available prior to 1880.

| Name of repository/database | Abbreviation | Region | $N_{all}$ | $N_{docu}$ | Reference | URL |
|---|---|---|---|---|---|---|
| PAGES2k proxy temperature database | PAGES2k | Global | 692 | 14 | Emile-Geay et al. (2017) | http://pastglobalchanges.org/science/wg/2k-network/data/ |
| NOAA/World Data Service for Paleoclimatology archives | NOAA Paleo | Global | >10'000 | 61 | | https://www.ncdc.noaa.gov/data-access/paleoclimatology-data |
| Euro-Climhist | Euro-Climhist | Switzerland/ Central Europe | 65 | 27 | Pfister et al. (2017) | https://www.euroclimhist.unibe.ch/en/ |
| Tambora.org | Tambora.org | Germany | 4 | 4 | Riemann et al. (2015) | http://www.tambora.org |
| National Snow and Ice Database: Global Lake and River Ice Phenology | NSIDC | Northern Hemisphere | 865 | 39 | Benson et al. (2000), updated 2020 | https://nsidc.org/data/G01377/versions/1 |
| Japan Climate Data Project | JCDP | Japan | 14 | 3 | | https://jcdp.jp |
| Climatological Database for the World's Oceans | CLIWOC | Global | 1624 | | García-Herrera et al. (2005) | https://www.historicalclimatology.com/cliwoc.html |
| Institute for Ocean Technology Ice Database | Ice Data | Canada | 4 | 4 | | http://www.icedata.ca |
| KNMI Climate Explorer | Climate Explorer | Global | | ~10 | | https://climexp.knmi.nl/start.cgi?id=someone@somewhere |
| Red Española de Reconstrucción Climática a Partir de Fuentes Documentales *Spanish Climate Reconstruction Network from Documentary Sources* | RECLIDO | Spain | 7 | 7 | | http://stream-ucm.es/RECLIDO/es/home-es.htm |
| Salvá Sinobas | Salvá Sinobas | Iberian Peninsula | 18 | 5 | | http://salva-sinobas.uvigo.es/index.php |
| Variabilidad y Reconstrucción del Clima | Vareclim | Global | 5 | 5 | | https://www.upo.es/vareclim/index.php |
| Reconstructed East Asian Climate Historical Encoded Series | REACHES | China | 1 | 1 | Wang et al. (2018) | https://www.ncdc.noaa.gov/paleo-search/study/23410 |
| Tracking Extremes of Meteorological Phenomena Experienced in Space and Time | TEMPEST | United Kingdom | 5 | 5 | Veale et al. (2017) | https://www.nottingham.ac.uk/research/groups/weather-extremes/research/tempest-database.aspx |

When it comes to global reviews of documentary evidence related to climate, only very few exist so far. Brázdil et al. (2018)
took on the task of compiling a global overview of studies on past droughts based on documentary evidence. Besides discussing
available source types and methods of documentary-based drought reconstructions, they provide an extensive compilation of
available documentary drought proxies from all continents. Similarly valuable is the global overview on climate indices in
climate reconstructions from archives of societies (Nash et al. 2021) that summarizes studies using climate indices to
reconstruct climate. They present a concise review of the many different derivation and verification practices for climate
indices based on qualitative documentary evidence and thus creating quantitative proxy data. In terms of marine data, the
climatological database on the world's oceans (CLIWOC) compiles documentary evidence from European ship logbooks



(García-Herrera et al., 2005). It contains observations from voyages for the pre-industrial period 1750-1854 from all over the world, which are transformed into quantitative data and thus available for climate reconstruction.

While documentary records relevant for climate reconstruction are available from all continents, continental-scale perspectives remain sparse, and no global-scale overview exists (Burgdorf, 2020). This article presents the first-ever comprehensive

inventory of globally available documentary evidence related to past climate and thus promotes a global perspective on historical documentary climate records. A strong focus lies on records that contain a significant potential for global-scale climate reconstructions. Therefore predominantly document-based climate time series are included. Many of the records included in this inventory are published in existing repositories. For these, merely directions where to find the data are given.

Following the introduction, in Sect. 2 structure, composition, and generation of the inventory are explained, and short

descriptions of datasets used to evaluate documentary data in the discussed case studies are given. In Sec. 3, I present the actual inventory and provide insights into the temporal and spatial availability of the data. To demonstrate the potential of documentary evidence for climate reconstruction three case studies from different periods are presented and compared with other datasets in Sect. 4. This is followed by some concluding remarks and perspectives upon future possibilities of documentary evidence for global climate reconstructions in Sect. 5.

## 2 Method and Data

### 2.1 Criteria for data collection

There is a great abundance of historical documentary evidence related to climate available from archives of societies and a considerable number of studies analyzing it. It was thus essential to define clear criteria before inventorying the data. First, only records published in predominantly English peer-reviewed journals, university theses, and official reports are considered.

Raw historical material is excluded to ensure that the records included in the inventory underwent thorough analysis and are interpreted in some way by qualified scholars a. Second, the period 1400-1880 is defined as the period of interest. Here, a balance is sought between data reliability and relevance as well as the possibility of calibration with other climate data like instrumental data series. Third, only climate information on temperature, precipitation, and wind is considered. Marine data from ship logbooks are generally excluded unless a time series for temperature, precipitation, or wind is derived thereof. A

further criterion is the length of the data series. Here I only include data with a minimum record length of 30 years (necessary for statistical analyses, e.g., allowing meaningful standardization), and out of those, a minimum of 20 years need to be before 1880 (otherwise, the value is questionable given the availability of observation-based data sets from that period onward). On top of these conditions, I started inventorying from the premise that the evidence at hand must show significant potential for climate reconstructions.





## 2.2 Structure

In the following, the structure of the inventory is presented. For each inventory entry, comprehensive information is available for the historical documentary evidence and the climate data that can be deduced from it. The information ranges from details on the historical sources, detailed meta-information, climate predictor, calibration and quantification of the data, to information

5 on the authors, corresponding references, as well as an indication as to whether the data is available. If the data is available, directions where one can find it is given. Table 2 provides an overview of the kind of information that is available for each entry.

**Table 2.** Overview of all the meta information that is available for every entry in the inventory.

| | | |
|---|---|---|
| **Climate parameter** | Nature of data at hand | p = precipitation, t = temperature, w = wind |
| **Resolution** | Temporal resolution | Monthly, seasonal, annual, decadal<br>if available, a specification of which season/month is given |
| **Time period** | - Start and end year of time series<br>- If available: start and end year of instrumental extension<br>- Length of series including gaps | |
| **Location** | - City/ Region<br>- Country<br>- Continent/ WMO region<br>- Elevation<br>- Specific locations: Lat, Lon<br>- Regions: Northernmost Lat, Southernmost Lat, Westernmost Lon, Easternmost Lon | Coordinates are either based on original values specified in corresponding publication or dataset, or when it comes to ranges, they are based on graphs in the corresponding publication. If no coordinates are given in the publication, google coordinates are used.<br>The column "Coordinates origin" indicate if they are original (=0), based on google (=1; here the associated location is given in column "Reference coordinates") or based on a graph (=2). |
| **Source Type** | Source type of the evidence at hand | Annals, chronicles, memoirs, inscriptions, personal diaries, legal-administrative or clerical evidence, scientific writing, letters, newspapers and journals, epigraphic evidence, weather compilations, ship logbooks, early instrumental observations |
| **Archives** | Archives where original sources are located | |
| **Type of Evidence** | Type of evidence serving as climate indicator | E.g., weather descriptions, reports of extreme events/famines/locusts/disease, tithes records, rogation ceremonies, phenological records (ice/plant phenology) etc. |
| **Indicators/Predictors** | Predictor/indicator from which climate signal is derived of | E.g., break- or freeze-up dates, grape/grain harvest dates, flowering dates, departure and arrival times, length of voyages, reports on various (extreme) events (droughts, snow, wind, rain and rainy season…), frequency of cold and warm years etc. |
| **Details** | Details on the derived climate proxy | E.g., information on the kind of index and how it is calculated |
| **Proxy** | Climate proxy | E.g., grape harvest dates, temperature/precipitation index, ice-severity index, Westwind index, flood catalog, cyclone chronology, monsoon starting date, reconstructed temperature/precipitation etc. |
| **Calibration** | Calibration of the data retrieved | Event catalogue (=e), time series (=ts), index (=i), reconstruction (=r), multi-proxy reconstruction (=r (multiproxy)) |
| **Quantification** | Quantification of the data retrieved | Direct observed/measured, indirect organic/inorganic (proxy data), cultural |
| **Contact** | E-mail address of corresponding author | |
| **Reference** | Relevant references | |
| **Data** | Indication of data availability and corresponding location | |
| **General comments** | Personal notes for the author | |



### 2.3 Data sets used for comparison

### 2.3.1 EKF400v2.0

EKF400 v2.0 (Valler et al., 2021) is a global, monthly, three-dimensional reconstruction based on an offline assimilation of early instrumental data, documentary data, and proxies (tree ring width, latewood density) into an initial condition ensemble

of 30 global model simulations using an ensemble Kalman filter assimilation technique (Bhend et al., 2012). The data set is given at a resolution of 2° and covers the period 1603–2004. The model is constrained, among other forcings, with annual sea-surface temperature reconstructions from Mann et al. (2008) to which interannual, El Niño–Southern Oscillation (ENSO)-related variability is added (see Bhend et al., 2012, for a method description). The data set is described in detail in Valler et al. (2021).

### 2.3.3 20CRv3

The 20CRv3 is the latest version of the Twentieth Century Reanalysis (20CR) Project and covers the period 1836 to 2015 (Slivinski et al., 2019). The atmospheric reanalysis dataset is based on the assimilation of surface pressure observations from the International Surface Pressure Databank (ISPD) version 4.7 (Compo et al., 2019) into the National Centers for Environmental Prediction (NCEP) Global Forecast System (GFS) model, version 14.0.1, using an 80-member ensemble

Kalman filter. The NCEP GFS model is run at T245 spectral truncation (corresponding to a horizontal resolution of 1° x 1°) and consists of 64 levels in the vertical. Two sea-surface temperatures (SST) products are used as boundary conditions: for the early period 1836-1980, the Simple Ocean Data Assimilation with sparse input, version 3 (SODAsi.3) (Giese et al., 2016); and for 1981-2015, the Hadley Centre Sea Ice and Sea Surface Temperature dataset, version 2.2 (HadISST2.2). In addition to that, the following forcings are prescribed: the solar forcing is based on the Total Solar Irradiance (TSI) Reconstruction by

Coddington et al. (2016), volcanic aerosols are based on Crowley and Unterman (2013), stratospheric ozone by Cionni et al. (2011) and atmospheric $CO_2$ levels by Saha et al. (2010).

### 3 Inventory

### 3.1. Global Overview

In the following, I present the inventory itself and highlight a selection of particularly relevant records. The inventory is

available as an electronic supplement online.

Data series included in the inventory can be categorized by the way they are calibrated. About half of the entries are time series of all different sorts (e.g., break-up and freeze-up dates, flowering and harvest dates, rain season lengths, rogation ceremony frequencies, flood and drought frequencies, voyage duration, etc.), followed by 36% indices (temperature, precipitation, Monsoon, dryness-wetness, wine quality, streamflow, ice severity or extend, etc.). Further categories are calibrated

reconstructions (11%), detailed event chronologies (2%), as well as very few multiproxy reconstructions (<1%).



Sources containing historical documentary evidence related to climate range from annals/chronicles, legal-administrative/ clerical documents, memoirs, personal diaries, travel reports to ship logbooks, epigraphic evidence, scientific writings, weather compilations, and newspaper articles. They can be divided into direct observations and indirect (proxy) data (Pfister et al., 1999). Direct observations include narrative reports on daily weather, climate anomalies, weather-induced hazards, and non-

weather-related events such as famines and epidemics. Indirect data are principally organic (plant phenology, information related to crop harvest) or non-organic (ice-snow phenology, information on water levels) in nature, but this category also includes cultural evidence (rogation ceremonies). 36% of the documentary evidence included in this inventory is based on direct observations, 38% on indirect inorganic (ice phenology) and 6% on indirect organic (plant phenology) proxy data, 2% of cultural proxies (rogation ceremonies), and 18% on a combination thereof. A small number of series are multi-proxy

reconstructions, combining documentary evidence with instrumental measurements and natural proxies.

It is important to note that direct observations are not necessarily more accurate than indirect proxy data: records of late flowering of cherry blossoms or delayed dates of river ice break-up can be much more precise than, for example, a record of someone noting a "cold winter".

In the inventory, a particular emphasis is given to ice-snow phenology due to its unique potential for reconstructing winter and

spring temperatures. While the Global Lake and River Ice Phenology database (Magnuson et al., 2000; Benson et al., 2000, updated 2020) includes freeze-up and break-up records from 865 lakes and rivers, only 39 of them are available before 1880 and have records longer than 30 years. Most of these records start around 1850, only 11 records extend as far back as 1830, and only five go beyond 1800. However, many more series exist and are now cataloged in this inventory. Particularly rich sources of information are two Russian publications dating back to the Russian Empire (Rykachev, 1886; Shostakovich, 1909).

These novel discoveries contain freeze-up and break-up records of over 1000 rivers and lakes from across the Russian Empire, many of which remained hitherto undiscovered by western scholars. The vast majority of these records are short, but 78 records feature a minimum length of 30 years prior to 1880 and are thus included in the inventory. 21 thereof go beyond 1800. In addition to these data from the Russian Empire, several novel river ice-phenology records from the north-eastern United States and Canada are included (e.g., Connecticut River (Barrat, 1840), Kennebec River (Gardiner, 1858), Mississippi River

(Shipman, 1938), Hudson River (Unknown Author, 1857), multiple locations in Hudson Bay (Catchpole et al., 1976; Magne, 1981; Ball, 1995), as well as many series from Europe (e.g., ice duration in the Haalem-Leiden Chanel (NL) (de Vries, 1977), first and last day of snowfall in Annecy (FR) (Mougin, 1912), river phenology from the Danube in Budapest (HT) (Takács et al., 2018), Lake Mälaren (SE) (Hildebrandsson, 1905; Eklund, 1999)). Furthermore, the inventory contains several sea ice records from the Baltic Sea (Koslowski and Glaser, 1999; Jevrejeva, 2001; Seinä and Palosuo, 2006), Iceland (Ogilvie, 1984;

Ogilvie and Jónsdóttir, 2000), the Hudson Bay area (Catchpole, 1995), the Hudson Strait (Catchpole and Faurer, 1983), the coast of Greenland (Schmith and Hansen, 2003),  Newfoundland (Hill and Jones, 1990), and the Labrador Sea (Teillet, 1988; Ouellet-Bernier and de Vernal, 2020).

Documentary evidence is available across all WMO Regions, is, however, unevenly distributed in space. While about half of all the time series stem from Europe (47%), an approximately even amount of evidence exists for Asia (18%), Africa (16%),





and North America (16%). South America and the South-West Pacific are sparsely covered (2% and 1% respectively) (Figure 1). For Europe and Asia, especially China, documentary series exist for both temperature and precipitation. For Africa and South America on the other hand, almost all the available records provide information on precipitation rather than temperature. The majority of records in the inventory are proxies for temperature (62%) (Fig. 1a) and precipitation (35%) (Fig. 1b), while only very few data series are available for wind (3%) (Fig. 1c).

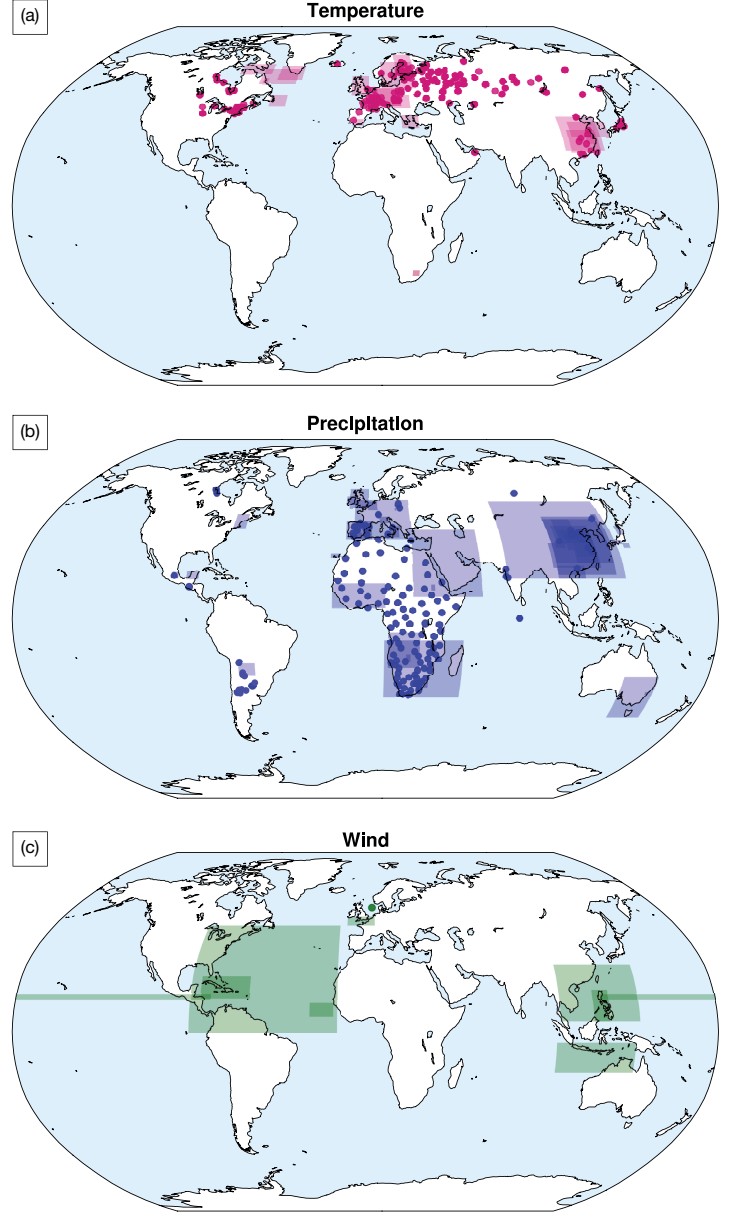

**Figure 1: Spatial distribution of available documentary series on climate from the global inventory for (a) temperature, (b) precipitation, and (c) wind. Dots indicate records assigned to a specific location; rectangles mark relevant domains for which climate information is found by a record.**



While some documentary series extend further into the past, beyond the Late Medieval Period, the focus here is on more recent evidence. Figure 2 depicts the temporal evolution of the available data series included in the inventory grouped into regions (WMO Regions). Only about 10% of the data series are available for the fifteenth century with most of the data being from Europe and Asia. The number of inventoried data series steadily increase throughout the sixteenth, seventeenth, and

eighteenth centuries across all regions and peak around the mid-nineteen century. This peak can be explained by the selection criterion to not include any series starting later than 1860. For the year 1849, a total of 577 series are included in the inventory. 45% thereof are series from Europe, 18% from Africa, 19% from Asia, and 16% are from North and Central America and the Caribbean. As for the entire time period, only very few records are available for South America and the South-West Pacific region adding up to about 2%. The majority of data series for Africa stems from a semi-quantitative

precipitation dataset covering 90 regions in Africa from 1801 to 1900 (Nicholson, 2001). The number of documentary series' is gradually decreasing in the second half of the nineteenth and twentieth century for all regions. The sharp drop in numbers for Europe around 1880 is due to the fact that many of the Russian ice phenology records published by Rykachev (1886) end around that time. The reason for the drop in the available numbers of Asian series at the beginning of the twentieth century is the fall of the Qing dynasty, Chinas last imperial dynasty, in 1911. The additional sharp drop in number in the twentieth

century coincides with the start of the Cold War in 1947. The overall drop in the number of series in the twentieth century can pose a complication since the overlap with instrumental series (needed for calibration) is often limited.

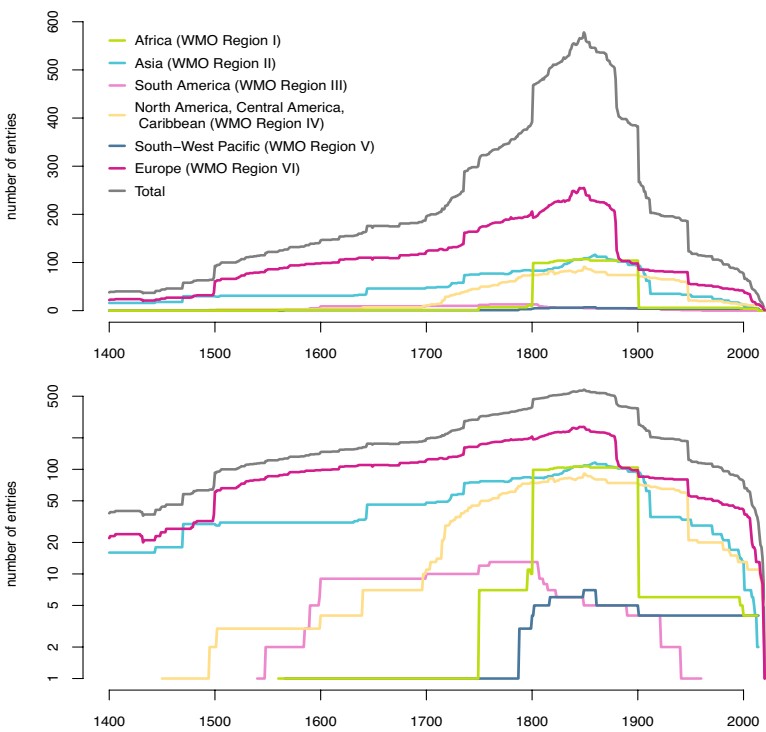

**Figure 2: Number of available documentary series related to climate as a function of time and region. Bottom panel depicts available series with logarithmic scaling of the y axis.**





### 3.2. Selection of relevant studies included in the inventory

In the following, a selection of relevant studies on documentary evidence related to climate from all WMO Regions is presented. The focus lies on studies with a large-scale or at least regional perspective. For a detailed overview of historical documents related to climate see the Palgrave Handbook of Climate History (White et al., 2018). It must be noted that

continental-scale overviews of documentary evidence with a global perspective remain sparse. Nonetheless, meta-information is available from all continents.

**Africa (WMO Region 1):** Most of the available African historical documents related to climate originated from a colonial context and largely consist of geographical studies, (meteorological) diaries, travel journals, and missionary reports. Thus, climate information from archives of societies is essentially limited to the nineteenth century. One prominent exception is the

Nile River flood series covering the period 622 to 1922 AD (Popper, 1951). Nicholson (2001) and Nicholson et al. (2012) provide a semi-quantitative, regional precipitation dataset combining historical documentary evidence and data from precipitation records into a seven-point scale index record for 90 geographical regions. This dataset, covering the nineteenth century, uniquely allows for an analysis of past rainfall variability for the African continent as a whole. On a regional and subcontinental scale, few overviews with a particular focus on seasonal rainfall variability are available predominantly for

Southern Africa (e.g., Neukom et al., 2014; Nash et al., 2016, 2019).

**Asia (WMO Region 2):** There exists an enormous abundance of historical documentary evidence for China spanning back many centuries. Chinese imperial dynasties have ruled the vast country and the many regional records, administrative and personal, about past climate and weather are well preserved (Ge et al., 2018). Maybe the most important historical record from China is the Compendium of Chinese Meteorological Records of the Last 3,000 Years edited by Zhang (1st Edition 2004 and

a revised and expanded 2nd Edition 2013). It covers a period starting in the sixteenth century BCE to 1911 and can be considered the world's most extensive compilation of written historical records related to climate (Wang et al., 2018). Further significant direct observed records are the Yearly Charts of Dryness/Wetness in China for 1470 to 1950 ( Chinese Meteorological Administration, 1981), as well as the Qing Yu Lu - Clear and Rain Records, providing daily records about the state of the sky, the wind and detailed information on precipitation events for 1724-1904 (e.g., Wang and Zhang, 1988). In

terms of indirect proxy evidence, the Yu-Xue-Fen-Cun Records of rainfall (i.e., Yu) infiltration and snowfall (i.e., Xue) during the Qing Dynasty (1644-1911) merit special emphasis. These quantitative records consist of measurements of soil infiltration depth and snow depth on the surface after a precipitation event in Chinese length units of Fen and Cun (e.g., Ge et al., 2005). These records, though technically measured (not observed), are included in this inventory as they are considered historical documentary evidence rather than instrumental measurements.

Due to the wealth of historical documentary evidence, it is possible to analyze past climate for large geographical areas far back in time, and thus many large-scale studies reconstructing past Chinese climate exist. Temperature and precipitation reconstructions at various spatial and temporal resolutions are available for several regions across China (e.g., Zhang, 1996; Wang et al., 1998, 2018; Song, 2000; Ge et al., 2005, 2010). Furthermore, there are many studies on particular regions: For eastern China (e.g., Wang and Wang, 1990; Zheng and Zheng, 1993; Zhou et al., 1994; Wang and Gong., 2000; Zheng et al.,



2001, 2006; Ge et al., 2005, 2008; Shen et al., 2008; Hao et al., 2012, 2016), southern China (e.g., Zhang, 1980; Zheng et al., 2012), as well as northern China (e.g., Zheng et al., 2005; Ge et al., 2011; Qian et al., 2012). Documentary data from eastern China are also used to reconstruct large-scale teleconnection patterns like the Arctic Oscillation (Chu et al., 2008), as well as the Pacific Decadal Oscillation (PDO) (Shen et al., 2006).

While many wide-reaching studies exist for China, very little are available for the rest of Asia, where historical climatology remains in its infancy. Some documentary evidence is available for Korea, but no historical documentary records related to climate were found for Thailand, Vietnam, Cambodia, Laos, or Myanmar. For the Indian subcontinent, historical sources related to climate predominantly become available with European colonialism. Studies on monsoon variability tap into documentary evidence but are limited to Western India (Adamson and Nash, 2013, 2014). Japan, on the other hand, possesses
written evidence on past climate extending back to the pre-Christian era. They are especially rich in ice- and plant-phenology proxy data that allows for a reconstruction of climate variability in Japan (Mikami, 2008). While individual proxy series are investigated in great depth e.g., the spring flowering of cheery trees in Kyoto (Aono and Omoto, 1994; Aono and Kazui, 2008; Aono and Saito, 2010), few studies combine their information and analyze past climate in a larger geographical context.

**South America (WMO Region 3):** Continental-scale compilations of historical documentary evidence are provided by
Neukom et al. (2009) and Prieto and García Herrera (2009). These compilations mainly contain documentary proxy data on precipitation from Argentina, Chile, Peru, Bolivia, as well as Ecuador extending back to the mid-sixteenth century. Furthermore, the two-volume collection of García Acosta (1996, 1997) assembles case studies covering 2000 years of Latin American climate history. While documentary evidence related to climate for Latin American exists beyond the European colonization of the Americas, the vast majority of written evidence stems from colonial times. A particularly great amount of
evidence was left by the Spanish Empire covering the entire period of their colonial presence in Latin America from the late fifteenth to the early nineteenth century (Prieto and Rojas, 2018).

**North America, Central America, and the Caribbean (WMO Region 4):** Historical climatology of North America is still in its early stages, overshadowed by the well-established field of climate reconstruction based on proxies from archives of nature. Nevertheless, some noteworthy sources and studies on evidence from the archives of societies exist. Their availability,
both spatially and temporally, is strongly linked to colonial settlement and thus predominantly limited to the East Coast and the Gulf of Mexico until the eighteenth century. See the concise review of White (2018) for an overview of the field. To date, North American historical climatology has predominantly focused on reviewing primary sources and individual records and extracting the available data. Only to a lesser degree was historical documentary evidence utilized for reconstructing past climate on a larger scale. Some studies on a more regional rather than local climate exist for the Hudson Bay area (Ball, 1995)
and New England (Baron et al., 1984; Baron, 1995; Baron and Smith, 1996). Furthermore, some sea ice severity reconstructions in Canadian waterways based on Hudson's Bay Company records are worth mentioning in this context (Catchpole et al., 1976; Catchpole and Faurer, 1983; Teillet, 1988; Catchpole and Hanuta, 1989; Catchpole, 1995).

In terms of Central American and the Caribbean, only few historical climatic records are scientifically evaluated. Besides large-scale studies focusing on Atlantic storm reconstructions (e.g., Chenoweth, 2006), some drought reconstructions for



Mexico as well as a precipitation reconstruction for the Pacific coast of Central America (Guatemala) (Guevara-Murua et al., 2018) exist.

**South-West Pacific (WMO Region 5):** Australian climate history is largely limited to European settlement starting in 1788 and is confined to the geographical area of south-eastern Australia (SEA) in early times (Gergis et al., 2018). Fenby and Gergis (2013) gathered the available historical documentary evidence and derived a dry-wet year chronology for the SEA region. This documentary chronology lies the foundation for the subsequently developed rainfall index that combines documentary evidence with early instrumental records (Gergis and Ashcroft, 2013; Ashcroft et al., 2014). The only historical documentary record for the South-West Pacific region included in the inventory, apart from Australia, exists for the Philippines, reconstructing the frequency of landfalling typhoons from the mid-sixteenth century onwards (García-Herrera et al., 2007; Ribera et al., 2008).

**Europe (WMO Region 6):** European historical climatology has a long tradition of exploiting the great wealth of diverse historical documentary evidence. For an extensive overview see Pfister et al. (2018) and Rohr et al. (2018). Depending on the geographical region, different kinds of records are utilized. Cultural evidence (rogation ceremonies) is predominantly available in the Mediterranean, plant phenology records in Western and Eastern Europe (especially France, Switzerland, Czech Republic), and ice phenology in North-Eastern Europe. Direct observed evidence (descriptions of daily weather and climatic and related environmental conditions (e.g., extreme events, famines, crop failures, etc.) is available with abundance in Central Europe. The plurality of the European records have local or regional foci. Studies on a larger geographical scale exist for the larger Mediterranean region (Camuffo et al., 2010), the Eastern Mediterranean and Middle East (Telelis, 2008; Xoplaki et al., 2018), the Meridional Balkans (Xoplaki et al., 2001), Central Europe (Glaser and Riemann, 2009; Dobrovolný et al., 2010), the Carpathian basin (Bartholy et al., 2004) and the Iberian Peninsula (Bullón, 2008; Rodrigo and Barriendos, 2008). While large-scale Europe-wide studies do exist, they are, for the most part, multi-proxy reconstructions (e.g., Luterbacher et al., 2004, 2016; Xoplaki et al., 2005; Pauling et al., 2006; Brázdil et al., 2010) that include a few documentary series. Since the historical information incorporated in these studies is only secondary, they are not included in the inventory. Despite the great wealth of documentary records available for Europe, no study to date has ever tried to combine these and analyze European climate on a continental scale based solely on historical documentary evidence.

**Russian Federation:** Although many historical sources describing past climate and weather exist for the Russian Federation, historical climatology remains largely marginalized, and reconstructions capitalizing on documentary evidence are scarce. The discipline is predominantly confined to Russian scholars. Borisenkov (1995) provides a review of relevant studies, mainly from the former Soviet Union east of the Ural Mountains. While not analyzed on the large scale yet, extensive compilations of ice phenology of water bodies from the Russian Empire dating back to 1530 (Daugava River, Riga) exist. Some of the therein included ice-break-up series like the Tornio River (Loader et al., 2011) or the Aura River (Norrgård and Helama, 2019) are well known and investigated. However, most of them are entirely novel discoveries with great potential for future research (Rykachev, 1886; Shostakovich, 1909).



The majority of record series included in this inventory are based on research since the 1990s. However, utilizing documentary evidence as climate predictors is not a new discipline in climate science. Many studies stem from the twentieth and even the late nineteenth century, and have since been forgotten, possibly because they missed the leap into the digital age (e.g., Brückner, 1890; Mougin, 1912; Shipman, 1938; Catchpole et al., 1976; de Vries, 1977).

**4 Case studies**

**4.1. Temperature anomalies after the Serua and unknown eruption 1693 and 1695**

To point out the value of documentary evidence for climate reconstructions, temperature anomalies after the strong tropical volcanic eruption of Mount Serua (Indonesia) in 1693 (Arfeuille et al., 2014) and a further unknown eruption in 1695 (Sigl et al., 2015) are analyzed. These two eruptions resulted in an unusual cooling over the Northern Hemisphere (NH), especially in

the summer months of the following years (Sigl et al., 2015). If this cooling is captured by natural proxies such as ice cores, one can assume that it must also have been documented in archives of societies. Especially relevant in this context are records related to harvest, which would have been impacted by cooling during the growing season.

To investigate a potential cooling signal in the NH following the 1693 and 1695 volcanic eruptions, standardized temperature signals from 52 available documentary records covering this timespan are analyzed. The temperature signal during the

anomalous period 1693-1697 (5 yr) is expressed relative to the combined average of the 10 years prior (1683-1692) and 10 years after (1698-1707). Each documentary series represents a temperature signal for a particular season or month. To compare these temperature composites, they are grouped into signals for spring (March-April), growing season (April-August), fall (October), winter (December-March), as well as an annual signal.

As shown in Figure 3a and b, 44 of the 52 documentary series show negative temperature anomalies (< -0.05 standard

deviations) for the years after the volcanic eruptions in 1693 and 1695. The signal is particularly homogenous over Europe, where the growing season during 1693-1697 was notably cooler than during the reference period. All but one of the European growing season proxies exhibit negative anomalies. They consist of temperature proxies based on various phenological parameters, e.g., grape and grain harvest dates, freezing of water bodies, duration of snow cover, as well as direct observations of the weather such as reports on temperature-related features (e.g., extreme frost periods). The most prominent anomaly can

be seen over the Carpathian basin, a proxy based on documentary evidence from Hungarian sources (Bartholy et al., 2004). Also, spring proxies in Europe (ice break-up on the Tornio River; Loader et al. (2011)) and Japan (dates of the cherry blossom; Aono and Omoto (1994), Aono and Kazui (2008), Aono and Saito (2010)) indicate cooler-than-normal conditions. For winter and fall, less evidence is available, and no clear signal emerges. The sole source for North America shows a strong negative anomaly based on the annual proxy in New England (Baron, 1995). This potentially indicates that colder-than-average

temperatures following the volcanic eruptions might not be restricted to the warm seasons but were rather a multi-annual event. The historical evidence relating to summer temperatures for China is based on the REACHES Climate Database (Wang et al., 2018), and the signal varies regionally.





**Figure 3: Temperature composites from documentary series for the 1693 and 1695 volcanic eruptions. Proxy series are categorized into seasonal groups: (a) shows spring and growing season proxies, whereas (b) comprises fall, winter, and annual proxies. (c) and (d) show composites of 2m surface air temperature during the growing season (April-August) and cold season (November-March) respectively based on the EKF400v2.0 reanalysis. Documentary proxies assimilated in the EKF400v2 are indicated by a black dot in (a) and (b).**





These findings based on documentary evidence correspond well with the temperature composite from the EKF400v2.0 reanalysis (Valler et al., 2021) for the spring and growing season (April-August) (Figure 3c) as well as the cold season (Figure 3d). One must note that 17 of the documentary series shown are assimilated in the EKF400v2.0 reanalysis (indicated with black dots), and thus, a good agreement is expected. The global reconstruction shows very strong negative anomalies over

Europe and indicates that the post-volcanic cooling after the 1693 and 1695 eruptions was especially strong over Europe. The temperature signal over Asia and North America is more ambiguous, particularly for the cold season. It should be noted that the cold season in the EKF400v2.0 reanalysis generally has little skill since it is predominately based on sea surface temperatures (SSTs) and the volcanic forcing.

### 4.2. Temperature anomalies after Laki eruption: June 1783 - February 1784

While for the late seventeenth century only relatively few documentary records are available, a more comprehensive picture can be drawn in terms of the temperature response to the Laki eruption in southern Iceland 1783-1784. While the Serua eruption in 1693 was a highly explosive, tropical eruption, the Laki eruption took place at a high latitude (64° N) and was much less explosive. However, the basaltic flood lava eruption lasted for eight months (8 June 1783 to 7 February 1784), and its consecutive eruptions maintained high atmospheric sulfur concentrations resulting in a persistent sulfuric aerosol cloud that

progressively covered large parts of the Northern Hemisphere for many months. The widespread pall of volcanic haze gave rise to considerable environmental implications and atmospheric effects (e.g., Sigurdsson, 1982; Brayshay and Grattan, 1999; Highwood and Stevenson, 2003; Thordarson and Self, 2003). For the period covering the Laki eruption, 90 documentary proxy time series are available. Due to their variety, the temperature response to the volcanic eruption can be analyzed on a seasonal resolution. Based on the available proxy series and the month they represent, seasons are defined as follows: Spring (February-

May), Growing Season (April-August), Fall (October-December), and Winter (December-March). Instrumental temperature records from 29 stations in Europe and North America show that the annual mean temperature in the three years following the Laki eruption was significantly colder relative to the 1768-1798 climatology (Thordarson and Self, 2003). The summer of 1783, on the other hand, was largely dominated by above-normal temperatures in Northern and Central Europe. To correctly capture the effect of the eruption lasting from June 1783 until February 1784, temperature composites are calculated

individually for each season. Anomalies are defined as the growing season of the year 1783, fall seasons of the years 1783-1784, and winter and spring seasons of the years 1784-1785. Similar to the 1690s, temperature responses during the relevant seasons are expressed relative to a reference period defined as a combined average of the 10 years prior and 10 years after. Composites are based on standardized values and are only formed if there are at least 16 reference years available in the documentary proxy series.

Firstly, a nice agreement across the seasons among the documentary proxy signals can be recognized. The growing season of 1783 is strongly influenced by the immediate effect of the volcanic aerosols. The proxies over Europe, in proximity to the Laki eruption, unanimously show a prominent warming signal (Figure 4a). The positive anomalies decrease towards the



Mediterranean Sea which is in very good agreement with instrumental records (Thordarson and Self, 2003). Particularly strong anomalies are found for the Low Countries (Shabalova and van Engelen, 2003). The strong warming signal over Central Europe is also captured by the EKF400v2.0 reanalysis (Figure 4e). That is not surprising since EKF400v2.0 assimilates some of the documentary proxy series (indicated by the black dots). As volcanic aerosol spread through the atmosphere, its secondary

and more wide-ranging cooling effect did establish in the course of the fall season of 1783. Fall documentary proxies show mixed signals suggesting that it was colder than average during 1783-1784 over North America but rather warm in northeastern Europe and Russia (Figure 4b). Advancing into winter and spring season, documentary proxies show very cold anomalies over central and northern Europe and North America (Figure 4c, d). Eight of the 18 proxies in Europe exhibit negative anomalies exceeding 1.5 standard deviations.

EKF400v2.0 agrees very well with these findings, confirming the strong negative anomalies over central and northern Europe and eastern North America (Figure 4g, h). Documentary proxies indicate that strong negative temperatures anomalies continue into spring all across the Northern Hemisphere. In general, we see a good agreement between the documentary proxies and EKF400v2.0, especially over Europe and North America. For China, both documentary data and EKF400v2.0 are rather uncertain, and we do not know if we can trust either. The signal in spring and fall over northeastern Europe and Russia differs

from EKF400v2.0, often showing rather opposites.





**Figure 4: Temperature composites from documentary series for the Laki eruption 1783-1784. Proxy series are categorized seasonally: (a) shows the growing season proxies for 1783, (b) fall proxies for 1783-1784, (c) winter proxies for 1784-1785, and (d) spring proxies for 1784-1785. Composites that are very close to 0 are colored grey, regardless of their sign. (e) through (h) show composites of 2m surface air temperature during the corresponding seasons based on the EKF400v2.0 reanalysis. Documentary proxies assimilated in the EKF400v2 are indicated by a black dot in (a) through (d).**





### 4.3. Precipitation anomalies during global drought 1877-1878

In the previous two case studies, two applications of documentary temperature proxies are presented. Besides temperature, historical documentary evidence on dryness and wetness are equally, if not more important, and provide invaluable insight into regions of the world where precipitation rather than temperature is the limiting factor for harvest

or general wellbeing. Here, I present a precipitation case study for Africa during the particularly dry period 1876-1878, sometimes referred to as the Global Drought (Brönnimann, 2015), or the late Victorian Great Drought (Davis, 2001). During this period, concurring prolonged droughts in South and East Asia, Africa, South America, and the Mediterranean region caused substantial crop failures, catalyzing some of the most extreme and widespread famines in Modern times (Davis, 2001; Cook et al., 2010; Singh et al., 2018). It is widely acknowledged that the record-breaking El Niño event

of 1877-1878 is the driving force behind these extraordinary drought conditions, affecting much of the tropics (Kripalani and Kulkarni, 1997; Aceituno et al., 2009; Brönnimann, 2015). However, Singh et al. (2018) suggest that an extraordinary combination of preceding cool tropical Pacific conditions in 1870-1876, a record strong Indian Ocean dipole (IOD) in 1877, a record warm North Atlantic Ocean in 1878, as well as the strong El Niño event 1877-1878 lead to these extreme drought events and the so-called Global Famine.

On the African continent, northeastern Africa, as well as the South, were particularly affected by drought conditions. That is also captured by historical documentary evidence. Figure 5a displays composites of precipitation proxies for the years 1877-1878 relative to a reference period defined as the combined average of the 10 years prior and 10 years after. The majority of the data points are based on the semi-quantitative precipitation data set from Nicholson (2001) and are annually resolved (rhombi). Another annual proxy is the Nile River flood series from Cairo (Popper, 1951). In addition

to that, five summer rainfall zone (SRZ) proxies (October-April) for Namibia (Grab and Zumthurm, 2018), Lesotho (Nash and Grab, 2010), as well as for the regions of Kalahari (Nash and Endfield, 2002, 2008), KwaZulu-Natal (Nash et al., 2016) and the Eastern Cape (Vogel, 1989) are depicted with squares. Furthermore, circles indicate winter rainfall zone (WRZ) (April-September) proxies for Namaqualand (Kelso and Vogel, 2007), as well as for the Eastern and the Southern Cape (Vogel, 1989).

The documentary proxies indicate strong to extreme drought signals for northeastern Africa, particularly Egypt and Sudan. The Nile River, for example, shows below normal water levels (Popper, 1951). For equatorial East Africa, on the other hand, documentary proxies indicate wetter than normal conditions. Strong positive precipitation anomalies in this region, particularly for the short rain season (September-November), are often associated with a warm ENSO episode, especially when coinciding with a positive IOD event (Black, 2005). An abundance of rain in East Africa is contrasted

by severe drought conditions affecting much of the SRZ in southern Africa. The most extreme drought signals are found for both the Eastern Cape SRZ and WRZ (Vogel, 1989). This dipole in precipitation is found to be associated with a positive IOD event occurring along with developing El Niño (Goddard and Graham, 1999). The southwestern Cape





region, where precipitation is dominated by austral winter rainfall, is characterized by predominantly positive anomalies, with the Namaqualand WRZ precipitation index showing the strongest anomaly (Kelso and Vogel, 2007). That suggests that for southern Africa, drought conditions were particularly severe during the austral summer months, and thus the regions in the SRZ were affected the most.

The temperature anomalies based on the two reanalyses EKF400v2.0 (Figure 5b) and 20CRv3 (Figure 5c) agree well on the annual drought signal in southeastern Africa, however, mostly miss the positive anomaly in the Southwest. Both EKF400v2.0 and 20CRv3 are predominantly driven by SSTs, hence a good agreement is expected. It is important to note that none of the presented documentary records are assimilated in EKF400v2.0 or 20CRv3. When looking into the seasonal differences between austral summer and winter precipitation, EKF400v2.0 somewhat captures the positive

signal for the winter months, whereas 20CRv3 shows positive anomalies on the eastern side of the Cape. While the rainfall deficits in the South are generally captured, both the positive anomalies in equatorial East Africa and negative anomalies in the Northeast are poorly represented. EKF400v2.0 shows the opposite signal for most of equatorial East Africa, and 20CRv3 misses out on the negative anomalies in the Northeast. It must be noted that precipitation oftentimes cannot be trusted in reanalysis products due to its large variability. Nevertheless, some largescale features are captured.

It becomes, however, evident that historical documentary evidence contains invaluable information that might help improve such products in the future.

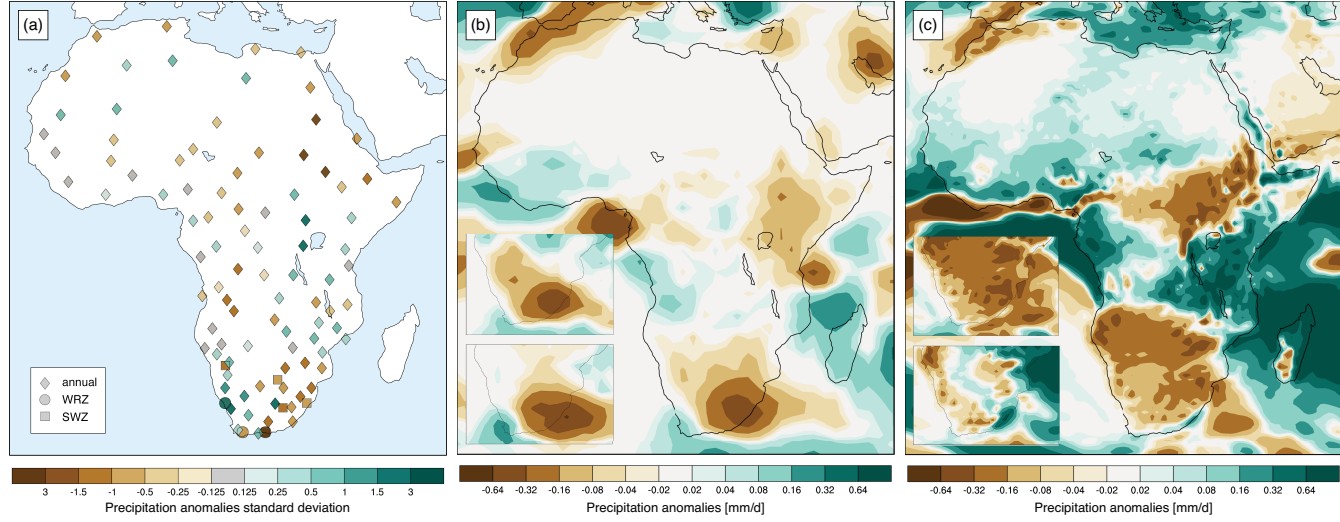

**Figure 5: Precipitation composites from documentary records for the Global Drought 1877-1878 (a). Composites that are very close to 0 are colored grey, regardless of their sign. Composites of annual accumulated precipitation are shown for EKF400v2.0 (b) and 20CRv3 (c). The superimposed graphs indicate composites for the SRZ (top) and the WRZ (bottom).**





# 6 Conclusion

This study presents the first-ever global inventory of historical documentary evidence related to climate and puts them in a large-scale context. It systematically compiles predominantly English research, for the most part, published in peer-reviewed journals that focuses on deriving climate information from documentary sources from archives of societies. The inventory
includes a great number of document-based climate time series with significant potential for large-scale climate reconstructions. The majority of series included are based on research since the 1990s. However, the inventory also contains numerous records from previous times that did not make the transition into the digital age and had to be rediscovered. Historical documentary evidence has largely been overlooked and has not received the appropriate consideration in the field of climate reconstruction. On one hand this might be because historical data is not readily available on a common platform or not
published altogether. On the other hand, they have never been considered in a global context until now. Although there is relatively little documentary evidence available compared to natural proxies or instrumental measurements, the historical information can be highly accurate and thus is of great significance. In contrast to natural proxies that are mainly annually resolved they generally exhibit a higher temporal resolution, often seasonal or even monthly. That is particularly valuable when reconstructing the boreal cold season that is poorly represented by natural proxies. European, Russian, Asian, and North
American ice- and plant-phenology records play an essential role in this regard as they facilitate high-resolution temperature information for October through May. Not only is there a great wealth of documentary time series' enabling temperature reconstructions, but there are also a considerable number of records available that serve as precipitation proxies through dry-wet indices. These might be of particular interest in arid subtropical regions (e.g., in the Mediterranean, China, and Africa), where other sources of information are sparse.
The presented case studies effectively demonstrate the significant potential documentary proxies alone have for reconstructing past temperature and precipitation variability on a continental and even hemispherical scale. This global documentary inventory of records related to climate compiles a set of essential documentary records from all continents and thus puts them in a large-scale perspective. That way, it creates the foundation for incorporating historical documentary evidence into climate reconstructions on a global scale.

**Acknowledgements**. This work was supported by Swiss National Science Foundation Project WeaR (188701), by the European Commission (ERC Grant PALAEO-RA, 787574), and by the Newton Fund/WCSSP South Africa. Simulations were performed at the Swiss National Supercomputing Centre CSCS. Support for the Twentieth Century Reanalysis Project version
3 dataset is provided by the U.S. Department of Energy, Office of Science Biological and Environmental Research (BER), by
30 the National Oceanic and Atmospheric Administration Climate Program Office, and by the NOAA Physical Sciences Laboratory.



**Special issue statement.** This article is part of the special issue "International methods and comparisons in climate reconstruction and impacts from archives of societies". It is not associated with a conference.

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
