# Peer review of "A global inventory of quantitative documentary evidence related to climate since the 15th century"

_Climate of the Past, 2021_

## Author Comment (AC1)

**REPLY TO THE REVIEWER COMMENTS**

(Reviewers' comments are in blue, replies on black, changes to the manuscript in green)

**REVIEWER #1**

Angela-Maria Burgdorf compiled an inventory of documentary evidence relating to climate history spanning the globe dating back to the fifteenth century of materials. In addition to her paper, she has also compiled a very impressive excel sheet, which can be found in the supplementary materials, with 686 data sets that she has analyzed for this paper.

Thank you for the careful revision and very constructive review of this paper. I appreciate the positive feedback and the helpful comments and suggestions. I have taken them up and revised the manuscript accordingly.

**General remarks**

Burgdorf's research presented here is impressive and provides an important and much needed first step of integrating the archives of society, i.e. documentary evidence in its many shapes and forms (diaries, ship log books, newspaper articles, clerical documents, chronicles, etc.) with the archives of nature, proxy data from tree rings, ice cores, etc. For the international community, this integration of data from the archives of society is very important as it is currently underrepresented and as historical records often include very precise information that can help the dating of extreme weather events or volcanic eruptions for instance. Furthermore what is also important is the global focus of this inventory, as very often the focus is on Europe, for which very good data exists, but it is crucial to also learn about the resources in other parts of the world and to use these. Her paper provides an inventory of those records that derive from mostly English-language sources that cover a period of at least two decades prior to 1880.

On page 5, line 21, the author mentions that "the period of 1400-1880 is defined as the period of interest." Here I would ask the author to elaborate on why she is specifically interested in this period. What makes this period so special? Why is it important to study and highlight this period? Why did the author pick 1400 as a start point and 1880 as the endpoint? As a historian, I have an idea why this time frame was chosen, but for the interdisciplinary audience that the journal aims at, I would like these to be spelled out further.

That is a good point and I happily add a sentence elaborating why this particular period was chosen:

"The period of 1400-1880 is defined as the period of interest since it marks a novel era in Western (European) history. The early modern period, following the Middle Ages, witnessed a break from medieval scholasticism and a surge of interest in Classical ideologies and values. The profound intellectual, social, and political transformations went hand in hand with continuous scientific evolution (e.g., the invention of the printing press in Europe in the 1450s) and European overseas expansion. This period is also characterized by novel approaches to observing and recording environment processes including the weather. As an effect, a great wealth of documentary material, both unpublished manuscripts, and early printed evidence exist in archives and libraries (Pfister et al., 2018). Moreover, an increasing number of (early) instrumental measurements essential for calibration are available from the seventeenth century onwards. However, with the continuous refinement of instrumental measurements practices and the establishment of national meteorological networks in the eighteenth and nineteenth century (Brönnimann et al., 2019), documentary evidence related to climate gradually became less relevant in many countries. Particularly after the launch of many national weather services around 1850 and the establishment of international measurement standards (Brönnimann et al., 2019), a marked decline in the research interest in non-instrumental climate records can be observed. Since this transition from non-instrumental to instrumental records did not occur simultaneously across the continents, 1850 would be too early as a cutoff year for some regions. Therefore, 1880 is chosen as the endpoint of this inventory. Another reason why I chose the period 1400-1880 is because many of the inventoried time series are assimilated in a novel climate reconstruction that starts in 1421 and solely includes instrumental measurements after 1880."

I want to compliment the author on the structure of her article, it made a lot of sense to introduce the inventory and relevant data before "testing" the inventory's use with three case studies, the first two case studies analyze volcanic eruptions, the 1693 Mount Serua (Indonesia) and unknown 1695 volcanic eruptions and the 1783 Laki eruption in Iceland. Here the author is able to contrast tropical (first case study) and high latitude eruptions (second case study). In this instance, I would have preferred more background information on the 1693 and 1695 eruptions, whether we know if the latter is tropical or not. (Perhaps the 2019 paper by Roseanne D'Arrigo et al. on "Complexity in Crisis: The volcanic cold pulse of the 1690s and the consequences of Scotland's failure to cope." In the Journal of Volcanology and Geothermal Research might be useful here). Considering at least the 1693 eruption was tropical, I wonder why the author chose to only look at the northern hemisphere in the maps of figure 3 and why not utilize the inventory to its full scale and draw a global picture, as far it is possible with the data available. The findings regarding the 1783 Laki eruption are in good agreement with current scholarship in the field of history. The third case study looked at precipitation anomalies during the global drought of 1877-1878, which was interesting and helpful to illustrate that the inventory is not only able to demonstrate temperature changes but also dryness/wetness. Considering that there are two examples of volcanic eruptions, I wonder if it might be useful to also show two droughts, perhaps a drought phenomenon in the early modern period, either global or regional.

Thank you for your inputs. I agree that it would be helpful to include some information on the two eruptions in the 1690s and to clearly state that the first example focusses on the climate response to a tropical eruption, whereas the latter demonstrates the climate response to a high-latitude eruption. The following section is added to the manuscript:

"To point out the value of documentary evidence for climate reconstructions, temperature anomalies for the unusually cold decade of the 1690s are analyzed. This particularly cold decade during the Little Ice Age was presumably forced by a series of volcanic eruptions, including the two significant tropical eruptions of Mount Serua (Indonesia) in 1693 (Arfeuille et al., 2014) and the even more potent unknown eruption in 1695 (Sigl et al., 2015; Toohey and Sigl, 2017). According to Sigl et al. (2015), the post-volcanic cooling in their aftermath gave rise to the 9th coldest decade (1692-1701) in Europe in the past 2000 years. It was of near-hemispheric scale and especially pronounced during the Northern Hemisphere (NH) summer months (Wilson et al., 2016). If this cooling is captured by natural proxies such as ice cores (e.g., Sigl et al., 2015) and tree rings (e.g., Wilson et.al., 2016; D'Arrigo et al., 2020), one can assume that it must also have been documented in archives of societies. Especially relevant in this context are records related to harvest, which would have been impacted by cooling of this scale during the growing season."

The reason this analysis is limited to the Northern Hemisphere is simply due to the fact, that there are no temperature records available for the Southern Hemisphere (see Fig. 1).

Thank you for the suggestion to add an additional drought case study. It would certainly be worthwhile to look at another example for a different region. However, considering that reviewer #2 believes the case studies to be redundant in the first place, I propose to stick with the three examples at hand.

The author shows a good knowledge of the field and relevant publications are cited throughout her paper. The number and quality of the references is appropriate. The author also clearly states her own original contribution in the paper.
The title clearly reflects the content of the paper and the abstract gives a good and complete summary of the paper.
The language and grammar of the paper are very good, fluent, concise, and easy to read.
I believe this paper would be a valuable contribution to Climate of the Past.

Thank you for your feedback, it is much appreciated.

**Specific remarks**

I found very few typos in the text, one such instance can be found on page 5, line 21, there is a random "… a." at the end of the sentence.

Thanks, that is not intended and the "a" is omitted in the revised manuscript.

I found the expression on page 10, line 1-2, "the focus here is on more recent evidence" too vague, I would ask the author to specify the time period she is interested in here, as there are scholars with vastly different research foci in Climate of the Past's audience (Holocene vs. the early modern / modern period).

Indeed, this expression is unspecific. I will change the sentence as follows in the revised manuscript:

"While some documentary series extend further into the past, beyond the Late Medieval Period, the focus here is on evidence from the Early and Late Modern Period."

Page 10: I did not understand why the numbers of these records appear to drop so sharply around 1880. This trend does seem very important, could the author elaborate on this change in data availability during this time? (Is it perhaps because of the author's focus on data that covers at least two decades before 1880?) It did not become clear to me what role the publications of Rykachev play in this context. I believe this needs a little bit more context for readers not familiar with this author.

Thank you for pointing out these ambiguities. I have added some context and explanations to the revised manuscript:

"The number of documentary series gradually decreases in the second half of the nineteenth and twentieth century for all regions. Partly this is simply because no new series are included after 1860. The reason for this is again the fact that many of the compiled series are used in a global climate reconstruction that after 1880 is solely based on instrumental measurements. More importantly, however, this correlates with the exponential development and expansion of instrumental measurement networks across the globe and a diminishing interest in non-instrumental records as a consequence thereof. The sharp drop in numbers for Europe around 1880 can be traced back to the availability of the ice phenology records from the Russian Empire published by Rykachev (1886). Out of these 119 record time series, 103 are not extended beyond Rykachevs publication from 1886. The large majority of these break-up and freeze-up series refer to rivers geographically located East of the Ural Mountains and are, thus, regarded as European series. They account for 85% of the European series ending between 1878 and 1882 and explain the sharp drop. Fourteen records from West of the Ural Mountains initially published by Rykachev (1886) are extended to the 1900s in the publication of Shostakovich (1909). The latter publication includes fifteen additional series from eastern Russia (regarded as Asian series) that also end in the 1900s. These records contribute to the visible drop in the available numbers of Asian series at the beginning of the twentieth century. A further reason for this drop is the fall of the Qing dynasty, China's last imperial dynasty, in 1911 since many Chinese documentary-based record series are based on the vast collection of institutional records from the imperial dynasties. There is an additional marked drop in the overall number of records in the twentieth century which coincides with the start of the Cold War in 1947. The gradual decline in numbers in the late nineteenth and twentieth century can pose a complication since the overlap with instrumental series (needed for calibration) is often limited."

On page 15, line 30, I found the expression "nice agreement" too colloquial.

Agreed, this is reworded in the revised manuscript:

"Firstly, we can recognize that the seasonal signals among the documentary records agree rather well, particularly for the growing and the winter season."

---

## Author Comment (AC2)

**REPLY TO THE REVIEWER COMMENTS**

(Reviewers' comments are in blue, replies in black, changes to the manuscript in red)

**REVIEWER #2**

The thematic focus of the article — a global inventory of "historical documentary evidence" — is highly relevant. And it is certainly desirable to create such an inventory. However, there are serious methodological issues to solve in order to create a useful and reliable inventory of this kind, and I doubt that these problems have been addressed appropriately in the reviewed version of the article. Methodolocal problems start with the problem of definition. In the context of historical climatology, "historical" and "documentary" evidence are synonymous. So, what is the meaning of "historical documentary evidence" in the first instance?

Thank you for that comment. When we talk about historical evidence, we can distinguish between historical instrumental (measured) evidence and historical documentary (written) evidence. In this inventory, the focus lays strictly on the latter, which is why I explicitly specified "historical documentary evidence" throughout the manuscript. However, I agree with the reviewer and will omit "historical" altogether starting with the title and throughout the manuscript to avoid redundancy. To specify that I am talking about historical documentary (written) evidence I adjust the sentence where "documentary evidence" appears for the first time in the following way:

"It combines information on past climate from all around the world, retrieved from many studies on documentary **(i.e., written)** sources. Historical evidence range from personal diaries, chronicles, administrative/ clerical documents to ship logbooks and newspaper articles."

Furthermore, I add the following sentence to the section where I define the criteria for the inventory:

"First and foremost, only written historical evidence are considered in this inventory. Other historical evidence such as (early) instrumental measurements are excluded since they are measured observations."

Moreover, the term "observations" is used in more than one way in this article, which requires clarification. In some of the earlier passages, observations are distinguished — at least implicitly — from the documentary record. In these passages, "observations" refer to measurements. Later, Pfister is quoted affirmatively for subsuming measurements under the broader distinction between "direct" and "indirect" information. Observations in his understanding — and I agree with him — not only include measurements, but also non-quantitative (in other words: qualitative) descriptions. These distinctions need clarification in order to produce a coherent explanation regarding the type of information gathered in the inventory.

Thank you for pointing out the lack of clarity around the term "observations". If I understand the reviewer correctly, he/she refers to the fact that observations can be used as a general term for "(early) instrumental measurements" as well as to refer to "direct observations" i.e., weather and phenological observations recorded in documentary evidence defined by Pfister et al. 1999.

The term "observations" is used ten times in the manuscript. In eight out of ten instances, the manuscript clearly states "which kind of" observation is meant e.g., "weather and phenological observations"; "early instrumental observations"; "the assimilation of surface pressure observations from (ISPD)"; "Sources containing historical documentary evidence related to climate […] can be divided into direct observations and indirect (proxy) data (Pfister et al., 1999)."; "Direct observations include narrative reports on daily weather, climate anomalies, weather-induced hazards, and non-weather-related events such as famines and epidemics".

There is, however, one instance in the manuscript where the term "observations" is used to refer to instrumental measurements which might be misleading (page 5, line 3). Therefore, I omit "observations" here and instead specify as follows:

"Here, I only include data with a minimum record length of 30 years (necessary for statistical analyses, e.g., allowing meaningful standardization). Out of those, a minimum of 20 years need to be before 1880, otherwise, the value is questionable given the availability of **instrumental measurement-based** data sets from that period onward)."

There is one additional instance where "observations" is not explicitly defined: "[CLIMWOC] contains observations from [ship] voyages for the pre-industrial period 1750-1854." From these ship voyages, both observations of the weather

(precipitation, clouds, winds) and early instrumental measurements are available. To be unequivocally clear that I am referring to the weather observations in this context I rephrase the sentence as follows:

"In terms of marine data, the climatological database on the world's oceans (CLIWOC) compiles documentary evidence from European ship logbooks (García-Herrera et al., 2005). It contains aside from temperature and air pressure measurements, **direct weather observations** (e.g., wind direction, wind force, present weather, sea conditions) from voyages for the pre-industrial period 1750-1854 from all over the world. The non-instrumental observations can be transformed into quantitative data and are thus available for climate reconstruction."

Moreover, I change the wording "instrumental observations" to "instrumental measurements" in all the above-mentioned instances to avoid misunderstandings.

The greatest challenge in creating a global inventory of non-quantitative documentary information, which is also much more difficult to solve, is related to language.

There are no qualitative records in this inventory. Every single record included is numerical which means that it is either an index, a time series of e.g., phenological dates, a calibrated temperature/precipitation reconstruction, or an event chronology. If the focus of the inventory were on qualitative documentary evidence, the issue related to language raised by the reviewer would be indeed very valid.

I agree, however, that fact that the inventory only includes quantitative records is not stressed out enough in the manuscript. To avoid misunderstandings in this context, I adjust the manuscript accordingly and emphasize throughout the article, that **only numerical/quantitative/derived records** are considered. For instance, I change the manuscript title and the abstract as follows:

"A global inventory of quantitative documentary evidence related to climate since the 15th century"

"Here, I attempt to compile the first-ever systematic global inventory of quantitative documentary evidence related to climate extending back to the Late Medieval Period."

The author is obviously not in a position, nor is it her ambition, to provide a survey of all available archival records related to climate history in the given timeframe (late middle ages to the present).

Again, this is a global inventory of relevant documentary climate data in **quantitative** form that only includes numerical record time series. Primary documentary sources are hence not included.

This is clear enough in the article. However, even a review of existing reconstructions based on the historical record (the instrumental record not counting) is almost impossible to achieve for a single person. It requires a consortium of authors capable of screening through older as well as more recent bodies of literature in multiple languages. I nevertheless believe that the findings presented in this article deserve attention and should be published. Yet, the problem of limits to the scope of such an inventory created by a single person should be problematized and addressed.

It would certainly be worthwhile to have a consortium of authors work on this topic. Clearly, the workload could have been carried out more efficiently, and the extent of the output would be even more comprehensive. Since no research group has yet taken on that task, my inventory can be thought of as a first step in assembling and connecting the vast number of quantitative documentary records in a global perspective which is largely missing to date. At best, it might stimulate further research in this direction. To acknowledge this limitation, I will add the following sentence to the conclusion:

"The inventory presented in this study contains a comprehensive set of highly relevant document-based climate time series, however, is certainly far from complete. It may serve as a starting point for further research and potentially stimulate a community effort to e.g., compile a full inventory of documentary evidence, including qualitative archival records"

Further problems: (1) The case studies in part 4 are unnecessary. Historical climatology is an established field of study. Hence, there is no need to demonstrate the value of historical climate information. It is unclear how part 4 relates to parts 1-3.

I disagree with the reviewer on this point. Admittedly, historical climatology is an established research field. However, as demonstrated in the manuscript, the vast majority of the relevant research and its analyses and applications to climate reconstructions is confined to a local or regional scale and is thus lacking a large-scale perspective. To my knowledge, no hemispheric case study using solely documentary records exists. Therefore, it is very relevant to point out the potential of combining individual records in such large-scale perspectives. Furthermore, documentary climate records are only very marginally considered in global climate reconstruction to date. The case studies strikingly underscore their potential for climate reconstruction and are thus essential in conveying the core message of this article. Reviewer #1 agrees with that and even suggests adding more case studies.

(2) The explanations given for the decline of number of historical records in the 19th (in some places) and 20th century (in other places) are too general and somewhat superficial. For example, the Chinese case — and even more so the Asian — is much more complex than can be grasped in one sentence stating that Imperial China ended in 1911. In this context it is particularly important that the author addresses the changing relation between the instrumental and the non-instrumental record.  (3) The English language requires considerable rephrasing and editing.

Thank you for this comment. Yes, I agree that my explanations for the decline of record numbers are too brief. In the revised manuscript I change the section to the following:

"The number of documentary series gradually decreases in the second half of the nineteenth and twentieth century for all regions. Partly this is simply because no new series are included after 1860. The reason for this is that many of the compiled series are used in a global climate reconstruction that after 1880 is solely based on instrumental measurements. More importantly, however, this correlates with the exponential development and expansion of instrumental measurement networks across the globe and a diminishing interest in non-instrumental records as a consequence thereof. The sharp drop in numbers for Europe around 1880 can be traced back to the availability of the ice phenology records from the Russian Empire published by Rykachev (1886). Out of these 119 record time series, 103 are not extended beyond Rykachevs publication from 1886. The large majority of these break-up and freeze-up series refer to rivers geographically located East of the Ural Mountains and are, thus, regarded as European series. They account for 85% of the European series ending between 1878 and 1882 and explain the sharp drop. Fourteen records from West of the Ural Mountains initially published by Rykachev (1886) are extended to the 1900s in the publication of Shostakovich (1909). The latter publication includes fifteen additional series from eastern Russia (regarded as Asian series) that also end in the 1900s. These records contribute to the visible drop in the available numbers of Asian series at the beginning of the twentieth century. A further reason for this drop is the fall of the Qing dynasty, China's last imperial dynasty, in 1911 since many Chinese documentary-based record series are based on the vast collection of institutional records from the imperial dynasties. There is an additional marked drop in the overall number of records in the twentieth century which coincides with the start of the Cold War in 1947. The gradual decline in numbers in the late nineteenth and twentieth century can pose a complication since the overlap with instrumental series (needed for calibration) is often limited."